# Characterization of genetic and molecular tools for studying the endogenous expression of *Lactate dehydrogenase* in *Drosophila melanogaster*

**Madhulika Rai**[1], **Sarah M. Carter**[1], **Shefali A. Shefali**[1], **Geetanjali Chawla**[2]*, **Jason M. Tennessen**[1]*

**1** Department of Biology, Indiana University, Bloomington, IN, United States of America, **2** Department of Life Sciences, School of Natural Sciences, Shiv Nadar Institute of Eminence (SNIoE), Dadri, Uttar Pradesh, India

* geetanjali.chawla@snu.edu.in (GC); jtenness@indiana.edu (JMT)

**Data Availability Statement:** All relevant data are within the manuscript and its Supporting Information files.

## Abstract

*Drosophila melanogaster* larval development relies on a specialized metabolic state that utilizes carbohydrates and other dietary nutrients to promote rapid growth. One unique feature of the larval metabolic program is that Lactate Dehydrogenase (Ldh) activity is highly elevated during this growth phase when compared to other stages of the fly life cycle, indicating that Ldh serves a key role in promoting juvenile development. Previous studies of larval Ldh activity have largely focused on the function of this enzyme at the whole animal level, however, *Ldh* expression varies significantly among larval tissues, raising the question of how this enzyme promotes tissue-specific growth programs. Here we characterize two transgene reporters and an antibody that can be used to study *Ldh* expression *in vivo*. We find that all three tools produce similar *Ldh* expression patterns. Moreover, these reagents demonstrate that the larval *Ldh* expression pattern is complex, suggesting the purpose of this enzyme varies across cell types. Overall, our studies validate a series of genetic and molecular reagents that can be used to study glycolytic metabolism in the fly.

## Introduction

Lactate dehydrogenase (Ldh) is a highly conserved enzyme that serves a key role in the regulation of cellular redox balance, glycolytic metabolism, and energy production [1, 2]. Although this enzyme has been studied for over a century [3, 4], we are still discovering new functions for both Ldh and lactate in metabolism, signal transduction, the cell cycle, and even gene expression [1, 2, 5–9]. Moreover, due to the central role that Ldh serves in cellular physiology, a wide variety of human diseases are associated with changes in Ldh expression and activity. For example, Ldh is well-known to play a central role in tumor metabolism, and enhanced Ldh serum levels are used as one of the diagnostic parameters in oral, laryngeal, melanoma, renal cell carcinoma and breast cancers [10–13]. The role of Ldh in human disease, however, extends far beyond cancer metabolism, as enhanced Ldh activity is also observed in diabetes

**Funding:** G.C is supported by the DBT/Wellcome Trust India Alliance Fellowship/Grant [grant number IA/I(S)/17/1/503085]. https://www.indiaalliance.org/fellowshiptype/basic-biomedical-research-fellowships. The funders had no role in study design, data collection and analysis, decision to publish, or preparation of the manuscript. J.M.T. is supported by the National Institute of General Medical Sciences of the National Institutes of Health under a R35 Maximizing Investigators' Research Award (MIRA; 1R35GM119557). https://nigms.nih.gov. The funders had no role in study design, data collection and analysis, decision to publish, or preparation of the manuscript.

**Competing interests:** The authors have declared that no competing interests exist.

and hyperglycemia [14–17], as well as during viral infections [18, 19]. In this regard, despite the well-documented relationship between elevated Ldh activity and human disease progression, questions remain about the tissue-specific mechanisms that link changes in Ldh expression and activity with the overall disruption of human health.

The fruit fly *Drosophila melanogaster* has emerged as an ideal system to study the role of Ldh in both healthy and diseased tissues. For example, Ldh activity levels fluctuate during the life-cycle in a predictable manner [20, 21]. In this regard, Ldh clearly plays a significant role during larval development, when the animal experiences a nearly 200-fold increase in body mass [22]. Ldh expression and activity are very high during this growth period when compared to other phases of fly life-cycle [20, 23, 24], indicating that lactate production is important for larval metabolism. Consistent with this hypothesis, *Ldh* mutations render larvae unable to maintain a normal redox balance, and when combined with *Gpdh1* mutations, induce significant growth defects [25]. In this manner, studies of Ldh provide an opportunity to understand how changes in Ldh expression affect *Drosophila* growth, development, and life-history events.

Similarly, studies of *Drosophila* disease models have also begun to focus on the link between lactate metabolism activity and tumor growth [for review, see 26], neuronal health and aging [27–34], as well as during infections and immune challenges [35]. Notable among these findings is that *Drosophila* tumors up-regulate Ldh in a manner that mimics the elevated Ldh-A activity observed in many human cancer cells [26, 36–40], indicating that Ldh in both flies and humans serves a beneficial role in tumorous growth. Thus, studies of *Drosophila* Ldh hold the potential to better understand how mammalian Ldh homologs function in development and disease.

Several genetic and molecular reagents have been used to examine *Drosophila* Ldh expression and activity. For example, early studies of Ldh in the fly relied on a biochemical enzymatic assay that generated an easily visualized staining pattern [24, 37]. These enzymatic assays have been supplemented with transgenes that rely on fluorescent proteins to either directly or indirectly visualize *Ldh* gene expression[35, 37, 39, 41]. However, most published *Ldh* reporters have not been directly compared, either with each other or with endogenous protein expression, raising the possibility that published genetic reagents might produce artifactual results. To address this issue and thus facilitate more precise studies of *Drosophila* Ldh function, here we compare the following genetic and molecular reagents: (i) a previously described *Ldh* genomic rescue construct that consists of a green fluorescent protein (GFP) coding region inserted immediately before the *Ldh* stop codon (referred to as *Ldh-GFP^Genomic*) [39], (ii) a previously described *Ldh-GFP* reporter that consists of *EGFP* inserted 50 bp upstream of the endogenous start site (referred to as *Ldh-GFP^enhancer*) [37, 42], and (iii) a commercially available *Drosophila* Ldh antibody (Bosterbio; DZ41222). As described below, we demonstrate that *Ldh-GFP^Genomic* produces a functional enzyme capable of rescuing the *Ldh* mutant phenotypes. Moreover, our studies reveal that all three reagents generate similar tissue-specific *Ldh* expression patterns. Interestingly, our analyses also reveal that Ldh is expressed in a complex manner during larval development, suggesting that this enzyme functions in multiple cell- and tissue-specific roles during larval development. Overall, our study enhances the ability of the *Drosophila* community to study Ldh within the context of both normal developmental as well as human disease models.

## Methods

### *Drosophila melanogaster* husbandry and genetic analysis

Fly stocks were maintained at 25˚C on Bloomington Drosophila Stock Center (BDSC) food. Larvae were raised and collected as previously described [43]. Briefly, 50 adult virgin females

and 25 males were placed into a mating bottle and embryos were collected for 4 hrs on a 35 mm molasses agar plate with a smear of yeast paste on the surface. Collected plates were stored inside an empty 60 mm plastic plate and placed in a 25˚C incubator.

*Ldh* mutations were maintained in trans to the balancer chromosome *TM3, p{Dfd-GMR-nvYFP}, Sb[1]* (BDSC Stock 23231). Unless noted, *Ldh* mutant larvae harbored a trans-heterozygous combination of *Ldh[16]* and *Ldh[17]* as described in previous studies [44].

Two transgenes were examined in our study. The *p{Ldh-GFP}* transgene was generated by inserting GFP immediately upstream of the *Ldh* stop codon within a previously described *Ldh* genomic rescue construct [39, 44]. For our analysis, the *p{Ldh-GFP}* was placed in the background of *Ldh* loss-of-function allele *Ldh[16]* [44]. The previously described *Ldh-GFP* enhancer trap line was a kind gift from Utpal Banerjee's lab [37, 42]. Finally, the *p{Ldh-mCherry}* transgene, which was previously described in a study of *Drosophila* hemocyte metabolism [35], is identical to the *p{Ldh-GFP}* except that the *mCherry* coding sequence was inserted immediately upstream of the *Ldh* stop codon. To distinguish between the two *Ldh-GFP* and the *Ldh-mCherry* constructs in the text, we will refer to the *p{Ldh-GFP}* and *p{Ldh-mCherry}* rescuing transgenes as *Ldh-GFP[Genomic]* and *Ldh-mCherry[Genomic]*, respectively, and the *Ldh-GFP* enhancer trap line will be referred to as *Ldh-GFP[Enhancer]*.

## Viability assay

Larval viability was measured by placing 20 synchronized embryos of each genotype on molasses agar plates with yeast paste and measuring time until pupariation. Wandering L3 larvae were subsequently transferred into a glass vial containing BDSC food and monitored until eclosion.

## Whole-larvae imaging

Expression of the *LDH-GFP[Genomic]* and *LDH-mCherry[Genomic]* transgenes in intact larvae and pupae was visualized without fixation using a MZ10F microscope with a EL6000 light source. GFP was visualized with the Leica ET GFP filter set (470 nm excitation filter; 525 emission filter). mCherry was visualized using the Leica ET mCherry filter set (560 nm excitation filter; 630 nm emission filter). Images were taken using a Leica MC170 HD microscope camera.

## Immunofluorescence

Larval tissues were dissected from mid-third instar larvae in 1X phosphate buffer saline (PBS; pH 7.0) and fixed with 4% paraformaldehyde in 1X PBS for 30 minutes at room temperature. Fixed samples were subsequently washed once with 1X PBS and twice with 0.3% PBT (1x PBS with Triton X-100) for 10 mins per wash.

For antibody staining of GFP in larvae expressing either *Ldh-GFP[genomic]* or *Ldh-GFP[Enhancer]*, tissues were dissected, fixed and incubated with goat serum blocking buffer (4% Goat Serum, 0.3% PBT) for one hour at RT and stained overnight at 4˚C with the primary antibody rabbit anti-GFP diluted 1:500 (#A11122 Thermo Fisher). Samples were washed three times using 0.3% PBT and stained with secondary antibody Alexa Fluor 488 Goat anti-Rabbit diluted 1:1000 (#R37116; Thermo Fisher) for either 4 hrs at room temperature or overnight at 4˚C. Stained tissues were washed with 0.3% PBT, immersed in DAPI (0.5μg/μl 1X PBS) for 30 mins and then mounted with vector shield with DAPI (Vector Laboratories; H-1200-10).

For larval tissues stained with the anti-Ldh antibody (Bosterbio; DZ41222), fixed tissues were washed three times with 0.03% PBT for 10 minutes and incubated in blocking buffer (3% bovine serum albumin) for 15 mins on a rocking shaker at room temperature. Tissues were then incubated with 200 μl of blocking buffer containing a 1:20 dilution of anti-Ldh on a

rocking shaker for 12 hrs at room temperature, 12 hrs at 4˚C, and a third incubation of 12 hrs at room temperature. Samples were then washed twice using 200 μl of blocking buffer with a 10 min room temperature incubation for each wash. Once the second wash was removed, 200 μl of blocking buffer containing a 1:1000 dilution of goat anti-rabbit Alexa 488 (Thermo Fisher; catalog #R37116) was added to the well and samples were incubated overnight on a rocking incubator at 4˚C. The next day, tissues were washed twice with blocking buffer, once with 0.03% PBT, once with PBS, and subsequently incubated for 15 mins in PBS containing 0.5 μg/ml of DAPI. Stained tissues were then mounted using vector shield containing DAPI (Vector Laboratories; H-1200-10).

The following primary antibodies from the Developmental Studies Hybridoma Bank (DSHB) were used to in our study: mouse anti-Prospero (MR1A-S, 1:500 dilution), rat anti-Elav (7E8A10, 1:500 dilution), mouse anti-Repo (8D12, 1:20 dilution). In addition, the mouse anti-Miranda antibody (1:20 dilution, [45]) was a kind gift from Alex Gould. Secondary antibodies used herein were goat anti-mouse Alexa Fluor 568 (Thermo Fisher, A11004; 1:1000 dilution) and goat anti-rat Alexa Fluor 568 (Thermo Fisher, A11077; 1:1000 dilution).

For all imaging studies, multiple Z-stacks of individual tissues were acquired using the Leica SP8 confocal microscope in the Light Microscopy Imaging Center at Indiana University, Bloomington and a representative section was used for the figures. The third instar larval CNS was imaged using a 20X objective, while all other tissues were imaged with a 40X objective. Excitation/emission max for Alexa Fluor 488 was 499/520 nm and for Alexa Fluor 568 was 579/603 nm.

## Gas Chromatography-Mass Spectrometry (GC-MS) analysis

Samples were collected, processed, and analyzed as previously described [46, 47]. For all experiments, six biological replicates containing 25 mid-L2 larvae were analyzed per genotype. GC-MS data was normalized based on sample mass and internal succinic-d4 acid standard.

## Statistical analysis of metabolite data

Statistical analysis was conducted using GraphPad Prism v9.1. Metabolic data are presented as scatter plots, with the error bars representing the standard deviation and the line in the middle representing the mean value. Data were compared using Kruskal–Wallis test followed by a Dunn's multiple comparison test.

## Results

### Genetic characterization of the *Ldh-GFP^Genomic* transgene

As a first step towards validating reagents for studying Ldh expression, we initially examined a GFP-tagged genomic rescue construct, referred to here as *Ldh-GFP^Genomic*, which has been previously used to study muscle development and imaginal discs tumors [39]. As described above and elsewhere, this transgene consists of a fragment of genomic DNA that contains the entire *Ldh* locus with GFP inserted at the 3' end of the coding sequence, immediately prior to the stop codon (Fig 1A). To determine if the *Ldh-GFP^Genomic* generates a functional GFP-tagged fusion protein, we assayed the ability of this transgene to rescue *Ldh* mutant phenotypes. Our genetic approach revealed that the resulting fusion protein appears functional, as the *Ldh-GFP^Genomic* transgene rescues both the lethal phenotype (Fig 1B and S1 Table) and metabolic defects displayed by *Ldh^16/17* mutant larvae (Fig 1C and 1D) [44]. We would note that *Ldh-GFP^Genomic; Ldh^16/17* mutant larvae exhibited slightly decreased levels of lactate and 2-hydroxy-glutarate (2HG) as compared with the wild-type control (Fig 1C and 1D and S1 Table),

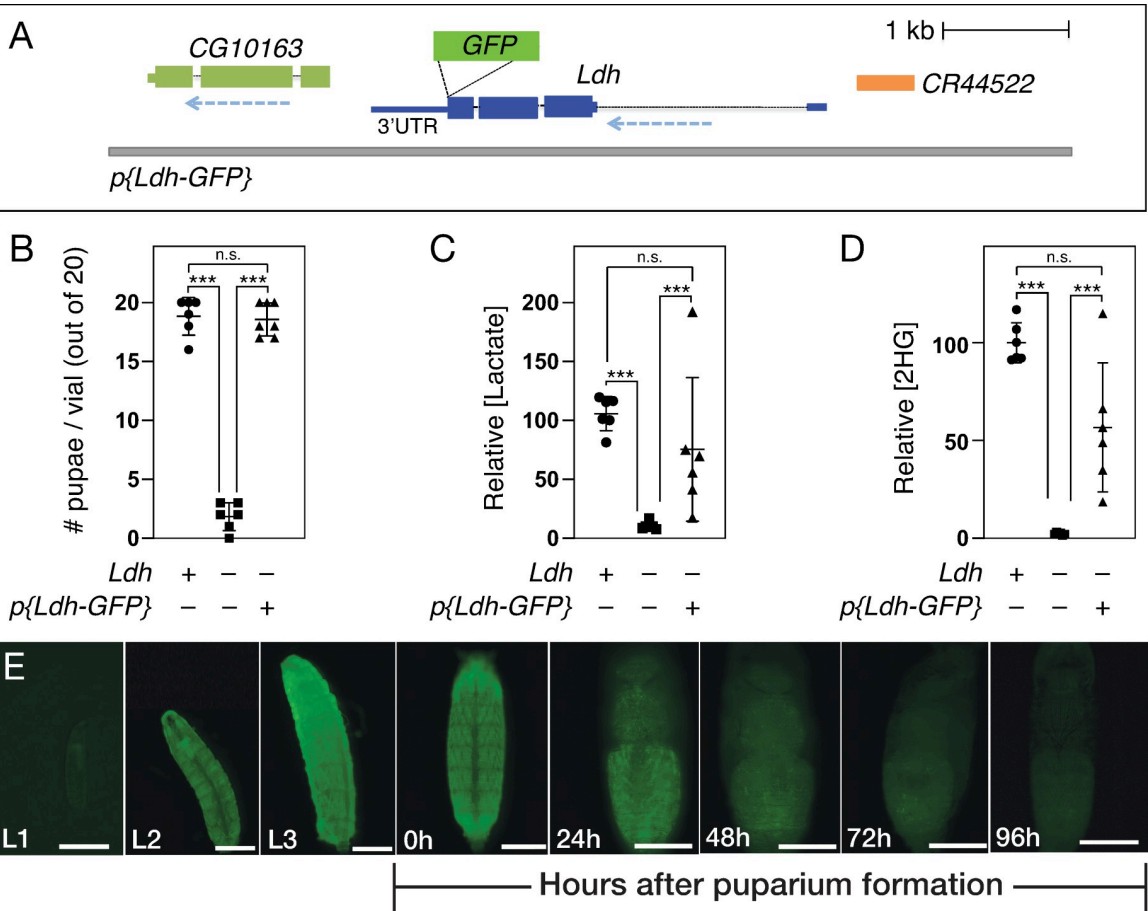

**Fig 1. Genetic characterization of the *p{Ldh-GFP}* transgene.** (A) A schematic diagram of the *Ldh-GFP^Genomic^* transgene illustrating the placement of GFP immediately before the *Ldh* stop codon. When this transgene is placed in a *Ldh^16/17^* mutant background, the resulting Ldh-GFP fusion protein is capable of rescuing (B) the *Ldh* mutant lethal phenotype and partially restoring (C) lactate and (D) 2-hydroxyglutarate (2HG) accumulation. (E) The *Ldh-GFP^Genomic^* temporal expression pattern is consistent with previous studies, with Ldh-GFP^Genomic^ being expressed at high levels throughout larval development and then declining at the onset of metamorphosis. Data in (B-D) analyzed using Kruskal–Wallis test followed by a Dunn's multiple comparison test. ***P<0.001.

however, the observed differences were not significant. Regardless, the general trend of decreased lactate and 2HG in *Ldh-GFP^Genomic^; Ldh^16/17^* mutant larvae suggests that the resulting fusion protein harbors lower activity than the endogenous enzyme. While there are a number of factors that could result in the Ldh-GFP enzyme exhibiting reduced activity relative to endogenous Ldh, a reasonable hypothesis stems from the fact that, in general, Ldh functions as a tetramer [48–50], thus the GFP tag could potentially disrupt the assembly or function of the higher order complex. Overall, our findings indicate that *Ldh-GFP^Genomic^* produces a functional Ldh enzyme.

In addition to assessing Ldh-GFP enzymatic function, we also assayed the gross expression pattern of *Ldh-GFP^Genomic^*. Previous studies have demonstrated that Ldh is highly active during larval development relative to other developmental stages–peaking during the L3 stage and gradually declining during metamorphosis [20, 51]. Consistent with previous observations, we noted that *Ldh-GFP^Genomic^* is expressed at such a high level during larval development that GFP was apparent in live larvae using a standard dissecting microscope. Overall, these whole animal expression levels mimicked previously reported changes in larval Ldh enzyme

expression and activity, with Ldh-GFP levels peaking in the L3 stage and declining thereafter. We would also note that *Ldh-GFP^Genomic* is so highly expressed in the larval central nervous system (CNS) and body wall muscles that GFP fluorescence from these tissues can easily be observed under low magnifications (Fig 1E). Overall, the observed *Ldh-GFP^Genomic* expression pattern is consistent with previous biochemical and genetic studies, as well as a recent manuscript that used this transgene to examine Ldh expression within body wall muscle [39].

In addition to the *Ldh-GFP^Genomic* transgene, we have also generated an identical version of this transgene labeled with mCherry, referred to herein as *Ldh-mCherry^Genomic*, which was previously used to study hemocyte metabolism [35]. In general, the *Ldh-GFP^Genomic* and *Ldh-mCherry^Genomic* transgenes exhibit similar spatial expression patterns, with notably high expression in the CNS and muscle (S1 Fig). However, the Ldh-mCherry fusion protein persists throughout much of metamorphosis (compare Fig 1E with S1C), suggesting that this fusion protein is either stabilized or accumulates to a higher level than Ldh-GFP. Since the *Ldh-GFP^Genomic* expression pattern more accurately reflects previously reported temporal changes in Ldh expression and activity, we chose to only characterize *Ldh-GFP^Genomic* in our subsequent experiments.

## Ldh is expressed in a complex pattern during larval development

Our analysis of the *Ldh-GFP^Genomic* transgene indicates that this genetic reagent can be used to reliably analyze *Ldh* expression. To further assess this possibility, we compared the tissue-specific larval expression pattern of the *Ldh-GFP^Genomic* transgene with *Ldh-GFP^enhancer*, as well as a previously undescribed *Drosophila* Ldh antibody (see methods; note that aLdh does not stain *Ldh^16/17* mutant tissues; S2 Fig). Our comparison demonstrated that all three reagents produced a similar cell- and tissue-specific larval staining pattern. Below we provide a brief description of the *Ldh* expression pattern in L3 larvae raised under standard growth conditions.

**Central nervous system.** The complexity of the *Ldh* expression pattern is perhaps most apparent in the central nervous system, where *Ldh* was highly expressed in the central brain and ventral nerve chord but absent from the optic lobe region (medulla, neuroepithelia and lamina; Fig 2A–2L). At the level of individual cell types, we observed identical expression patterns using both GFP transgenes as well as the Ldh antibody—*Ldh* is highly expressed in a subset of neurons and glia (Fig 3I–3P), as determined by co-staining with antibodies that recognize the neuronal protein Elav and the glial protein Repo [52, 53]. We would also note that *Ldh* expression was absent in neuroblasts that co-stain with a Miranda antibody [54, 55], but present in ganglion mother cells (GMCs) co-stained with Prospero antibody [54, 55]. Thus, our observations reveal that *Ldh* is expressed in a subset of GMCs (Fig 3E–3H), neurons (Fig 3I–3L) and glia (Fig 3M–3P), but not in neuroblasts (Fig 3A–3D). We are unsure as to the significance of this *Ldh* expression profile.

**Digestive and renal systems.** Both the larval gut and Malpighian tubules express Ldh in cell-specific manner. Within the gut, *Ldh* was expressed at relatively high levels in subset of cells within the adult midgut progenitors (AMPs) clusters, but at low or undetectable levels within larger EC cells (Fig 4A–4F). Considering that AMPs are small clusters of proliferating cells found within the larval gut epithelium [56], future studies should examine how Ldh influences the identify and proliferative capacity of these cells.

In the Malpighian tubules, Ldh levels were notably high in stellate cells as compared with principal cells and other renal cell types (Fig 4G–4L). This result is notable because stellate cells possess fewer mitochondria than principal cells [57]. Thus, our observation supports previous studies and suggest that stellate cells are more reliant on glycolysis than principal cells.

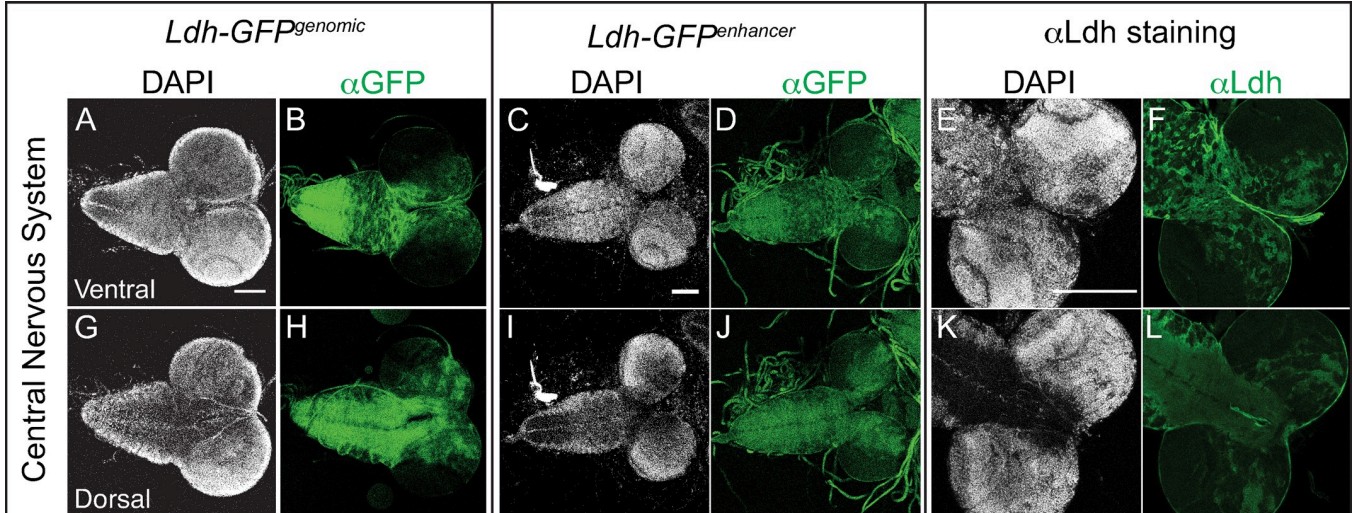

**Fig 2. Characterization of *Ldh* expression in the central nervous system.** The *Ldh* expression pattern in the central nervous system of L3 larvae was examined using *Ldh-GFP^Genomic^*, *Ldh-GFP^enhancer^*, and an aLdh antibody. (A-L) Representative confocal images of Ldh expression and DAPI staining in the (A-F) ventral and (G-L) dorsal sides of the CNS. The scale bar in all images represents 40 μM. Scale bar in (A) applies to (B,G,H). Scale bar in (C) applies to (D,I,J). Scale bar in (E) applies to (F,K,L).

**Fat body and salivary glands.** Previous studies indicate that *Ldh* gene expression as well as Ldh enzymatic activity are relatively low in the larval salivary glands and fat body [20, 58]. Consistent with this observation, Ldh expression was below the level of detection within both the fat body (Fig 5A–5F) and salivary glands (Fig 5G–5L).

**Imaginal discs.** *Ldh* expression has been extensively studied in the larval imaginal discs [26, 36–40]. These studies have consistently observed relatively low levels of *Ldh* expression during normal imaginal disc development. Our findings confirm earlier studies and again demonstrate that *Ldh* is present at a relatively low level within the leg, wing, and eye-antennal imaginal discs from L3 larvae (Fig 6A–6L). However, we found that *Ldh* is noticeably expressed within a few cells of these tissues. Patches of GFP expression were observed in the leg disc, which based on the similarity of this expression with that of *sens* [59, 60], we hypothesize to be the sensory organ precursors (SOPs). Also, both transgenic constructs resulted in GFP expression in cells of the eye-antennal disc posterior to the morphogenetic furrow (Fig 6E, 6F, 6K and 6L).

Overall, the similarities in expression between the *Ldh-GFP^Genomic^* transgene, the *Ldh-GFP^enhancer^* transgene, and the aLdh antibody in the examined larval tissues suggests that the *Drosophila* metabolism community can use any of these three reagents to study *Ldh*. However, we would recommend that any future study uses more than one of these reagents to validate observed changes in *Ldh* expression.

## Discussion

Here we demonstrate that two transgenes and a commercially available antibody reveal similar *Ldh* expression signatures during larval development. While the *Ldh-GFP^Genomic^* and *Ldh-GFP^enhancer^* transgenes were previously described and used for a variety of studies, our analysis indicates that either reagent can be used to reliably study *Ldh* expression. Moreover, our characterization of the *Ldh* antibody provides the first direct visualization of Ldh protein within the fly and will significantly enhance future studies of this enzyme. We would also note that while we did not fully characterize the *Ldh-mCherry* transgene in this study, the use of the

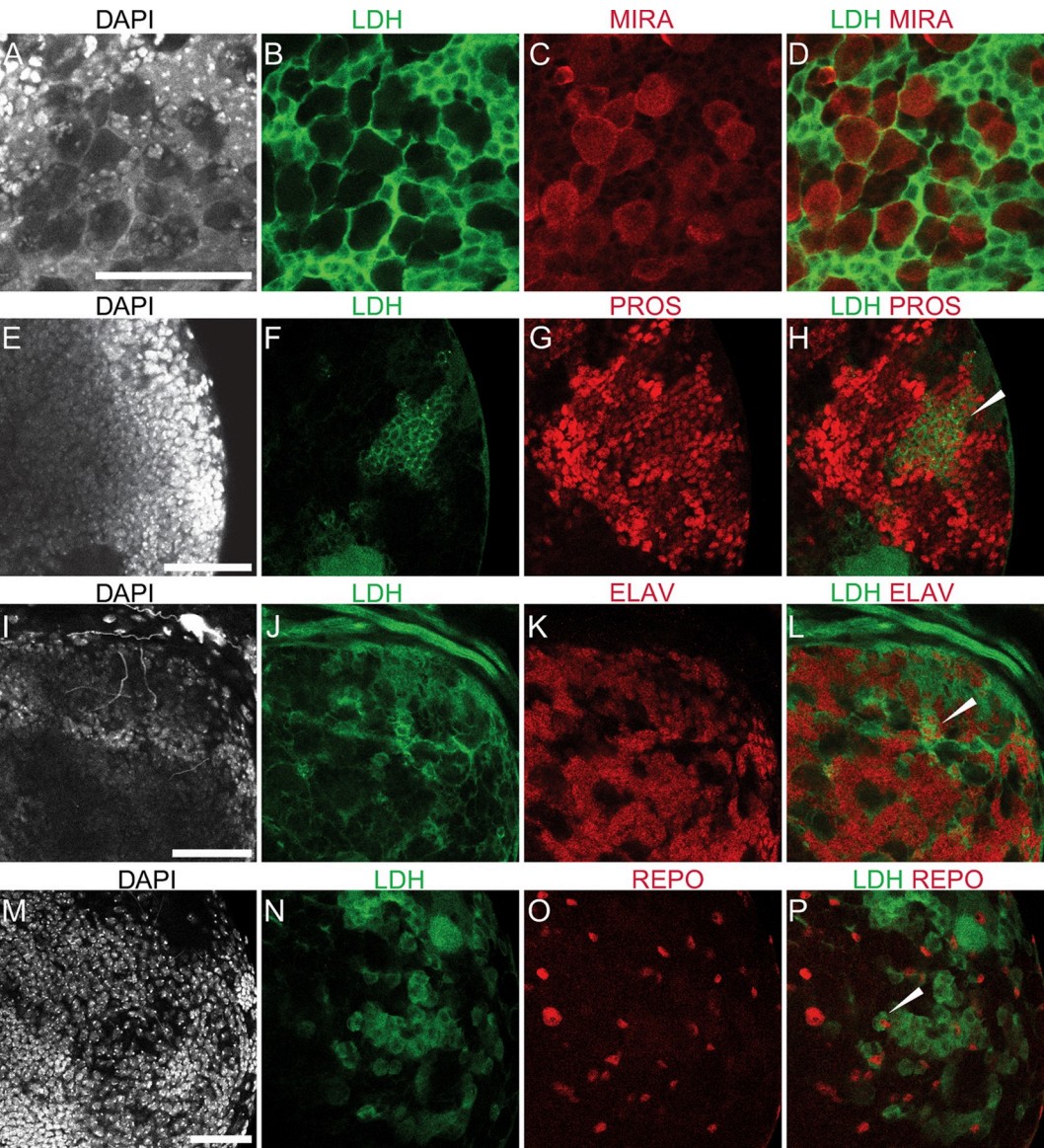

**Fig 3.** ***Ldh* expression in ganglion mother cells, neurons, and glia.** Representative confocal images showing the absence of *Ldh* in neuroblasts of third instar *Ldh-GFP^genomic* larvae. The first two columns from left to right show DAPI and Ldh. The last column shows the colocalization of *Ldh* with CNS cell specific markers. (A-D) *Ldh* is not expressed in neuroblasts, which are labelled with aMiranda (MIRA) antibody. In contrast, (E-H) *Ldh* is expressed in ganglion mother cells (GMCs), which are labelled with aProspero (PROS) antibody. Ldh-GFP is also present in both (I-L) neurons stained with aElav and (M-P) and glia stained with aRepo. The scale bar in all images represents 40 μM. Scale bar in (A) applies to B-D). Scale bar in (E) applies to (F-H). Scale bar in (I) applies to (J-L). Scale bar in (M) applies to (N-P). The scale bar in all images represents 40 μM.

mCherry fluorophore provides some flexibility in systems that are using GFP transgenes for other purposes. However, care should be used when using this reagent, as the resulting fusion protein appears to be stabilized relative to the endogenous enzyme. Overall, our study thus validates use of these reagents for future use by the *Drosophila* research community.

Beyond our initial validation of these three reagents, our study also reveals that *Ldh* is expressed in a complex pattern across tissues and cell-types. At the gross tissue level, our analysis agrees with studies dating back to the 1970s [20], as well as modern gene expression analyses that described how *Ldh* expression varies in intensity across larval tissues [35, 37, 39, 41,

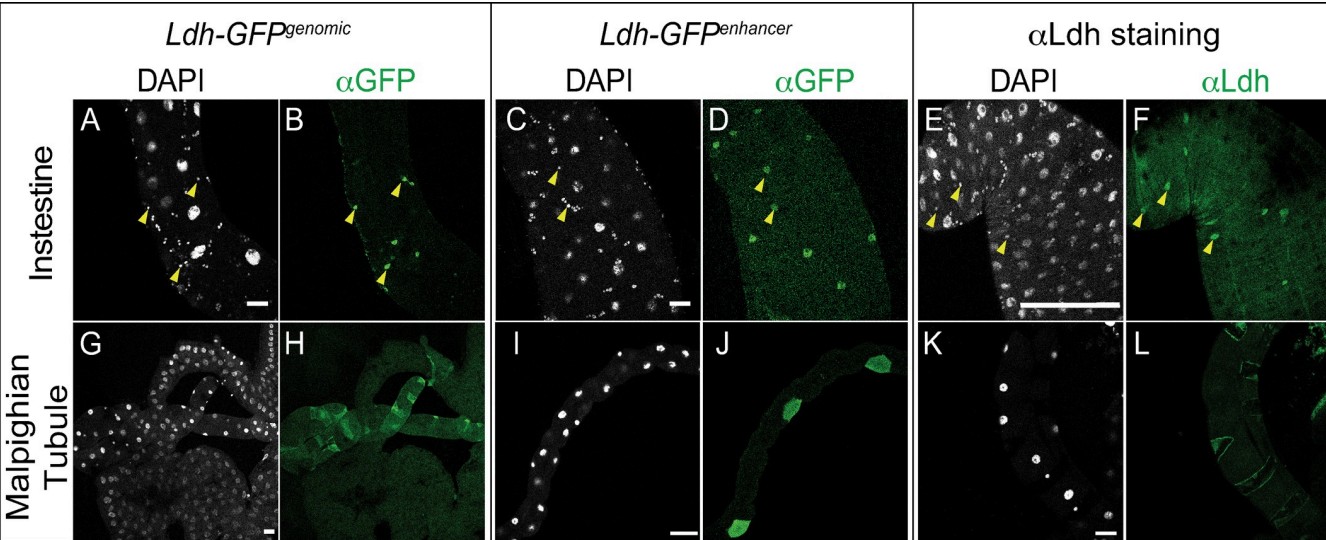

**Fig 4. Characterization of *Ldh* expression in the intestine and Malpighian tubules.** The *Ldh* expression pattern in the intestine and Malphigian tubules of L3 larvae was examined using *Ldh-GFP^Genomic*, *Ldh-GFP^enhancer*, and an aLdh antibody. (A-L) Representative confocal images of *Ldh* expression and DAPI staining in (A-F) the midgut and (G-L) the Malphigian tubules. Note that *Ldh* is expressed at significantly higher levels in (A,C,E) a subset of small cells of the AMP clusters (denoted with arrowheads in (A-F)) and (G,I,K) the stellate cells of the Malpighian tubules. The scale bar in all images represents 40 μM.

61]. Consistent with those early studies, we observe very high Ldh expression in muscle and relatively low or undetectable levels in the fat body and salivary glands. All three observations are interesting in regard to the metabolism of those tissues. For example, while high levels of *Ldh* expression in larval muscle would be expected based on the role of this enzyme in a wide variety of other animals, adults exhibit relatively low levels of *Ldh* activity [20], indicating that *Drosophila* muscle has evolved to meet the extreme energetic demands without using this enzyme and raising the question as to why there is such a dramatic difference in *Ldh* expression levels between larval and adult muscle. One likely explanation is that *Ldh* in the muscle, as

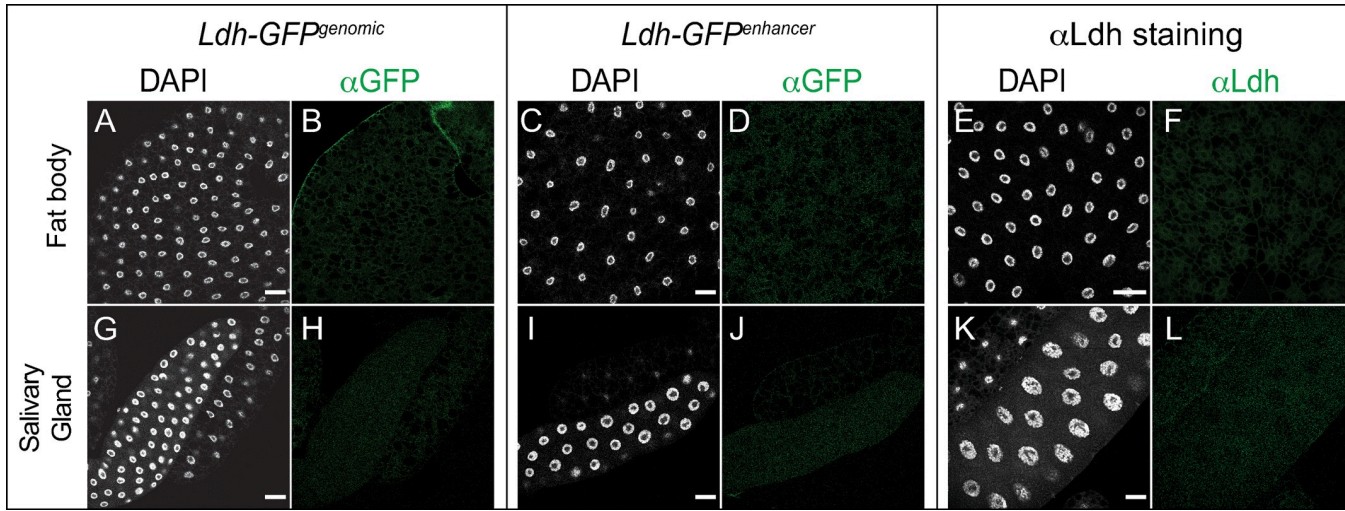

**Fig 5. *Ldh* expression pattern in the fat body and salivary gland.** The *Ldh* expression patterns in the fat body and salivary glands of L3 larvae were examined using *Ldh-GFP^Genomic* and *Ldh-GFP^enhancer*. (A-L) Representative confocal images of *Ldh* expression and DAPI staining in (A-F) the fat body and (G-L) the salivary glands. The scale bar in all images represents 40 μM.

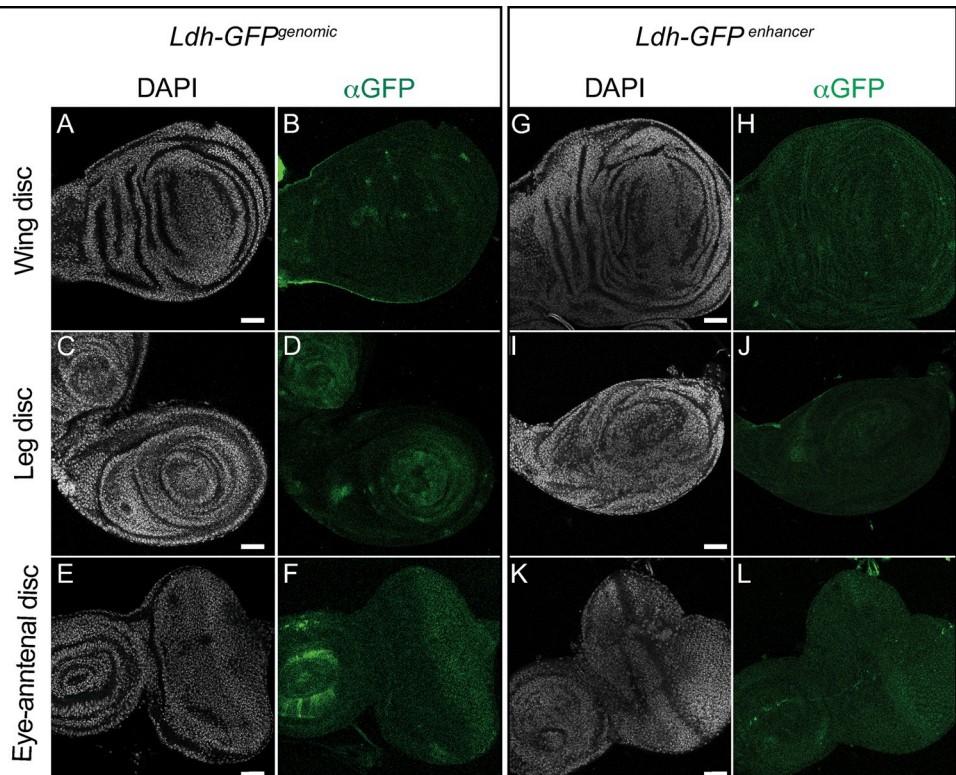

**Fig 6.** *Ldh* **expression pattern in imaginal discs.** Representative confocal images of imaginal discs (top to bottom: wing disc, leg disc and eye disc) dissected from third instar (D5 AEL) *Ldh-GFP* <sup>genomic</sup> (A-F) and *Ldh-GFP*<sup>enhancer</sup> (G-L) third instar larvae. DAPI is shown in white and *Ldh* expression is shown in green. Note that the aLdh antibody failed to produce any staining in the imaginal discs and thus was excluded from the figure. The scale bar in all images represents 40 μM.

well as other larval tissues, serves to buffer mitochondrial metabolism against the hypoxic environments often encountered by Dipteran larvae–a hypothesis supported by earlier studies [62, 63]. Future studies should both examine this possibility and investigate why larval salivary glands and fat body exhibit relatively low *Ldh* activity levels.

In contrast to the muscle, fat body, and salivary glands, other larval tissues display a complex expression pattern. For example, the AMP clusters of the larval midgut and stellate cells in the Malphigian tubules exhibit notably high levels of Ldh expression relative to other cell types in the surrounding tissue. The larval CNS, however, exhibits the most dramatic example of how Ldh expression can vary across cell types. Notably, there is a striking lack of *Ldh* expression in neuroblasts relative to other cell-types. Moreover, while we observe *Ldh* in both neurons and glia–a result consistent with the current hypothesis that lactate functions in a metabolic shuttle between these two cells types [27, 29]. However, we observed an unexpected heterogeneity in expression levels throughout the brain, with regions of high Ldh expression being observed adjacent to low levels. We are uncertain as to the significance of this observation.

Finally, we would highlight the distinct lack of *Ldh* in larval imaginal discs at the timepoint examined. Our observation is consistent with several previous studies, which describe how *Ldh* is expressed at low levels in normal imaginal discs but at dramatically higher levels in tumorous discs [26, 36–40]. Together, these suggest that the presence or absence of *Ldh* within the discs are of importance for cell growth and development. We would also note that much

earlier studies supported a model in which *Ldh* expression is somehow linked to imaginal disc development. After all, the original name for *Ldh* in *Drosophila melanogaster* was *Imaginal disc membrane protein L3* (*ImpL3*) [24, 64], which was identified as being induced in response to 20-hydroxyedcysone signaling. Consistent with this possibility, imaginal discs exposed to 20-hydroxyecdysone in culture exhibit a significant increase in *Ldh* expression [24].

In conclusion, our study validates the use of both genetic and molecular reagents to accurately study *Ldh* expression during larval development. Moving forward, we would encourage the fly community to use these reagents in combination when conducting studies of *Ldh* expression.

## Supporting information

**S1 Fig. Expression of *Ldh-mCherry*^Genomic during larval development.** The *Ldh-mCherry*^Genomic spatial expression pattern is consistent with previous studies, with Ldh-mCherry^Genomic being expressed at high levels in the body wall muscle. However, unlike Ldh-GFP^Genomic, the expression of mCherry^Genomic fusion protein persists throughout much of pupal development (compare with Fig 1B).
(PDF)

**S2 Fig. aLdh immunostaining in control and *Ldh* mutant larval tissues.** Representative confocal images of the (A-H) central nervous system, (E-H) intestine, (I-L) fat body, and (M-P) salivary gland dissected from third instar $w^{1118}$ and $Ldh^{16}$ larvae. DAPI is shown in white and *Ldh* expression is shown in green. Note the brightness in (L) was increased by 30% to highlight the lack of staining in small cells within AMP clusters. The scale bar in all images represents 40 µM. The scale bar in (A) applies to panels (B-H), the scale bar in (I) applies to panels (J-L), and the scale bar in (M) applies to panels (N-P).
(PDF)

**S1 Table. Data for Fig 1B–1D.** Data and statistical analysis used to generated figure panels 1B-1D.
(XLSX)

## Acknowledgments

We thank Alex Gould for generously sharing the Miranda antibody. The antibodies recognizing Elav, Repo, and Prospero were obtained from the Developmental Studies Hybridoma Bank, created by the NICHD of the NIH and maintained at The University of Iowa, Department of Biology, Iowa City, IA 52242.

## Author Contributions

**Conceptualization:** Madhulika Rai, Sarah M. Carter, Shefali A. Shefali, Geetanjali Chawla, Jason M. Tennessen.

**Data curation:** Madhulika Rai, Sarah M. Carter, Shefali A. Shefali, Geetanjali Chawla, Jason M. Tennessen.

**Formal analysis:** Madhulika Rai, Shefali A. Shefali, Geetanjali Chawla, Jason M. Tennessen.

**Funding acquisition:** Geetanjali Chawla, Jason M. Tennessen.

**Investigation:** Madhulika Rai, Geetanjali Chawla, Jason M. Tennessen.

**Methodology:** Madhulika Rai, Sarah M. Carter, Geetanjali Chawla, Jason M. Tennessen.

**Project administration:** Geetanjali Chawla, Jason M. Tennessen.

**Resources:** Geetanjali Chawla, Jason M. Tennessen.

**Supervision:** Madhulika Rai, Geetanjali Chawla, Jason M. Tennessen.

**Validation:** Sarah M. Carter, Shefali A. Shefali, Geetanjali Chawla.

**Visualization:** Sarah M. Carter, Shefali A. Shefali, Geetanjali Chawla, Jason M. Tennessen.

**Writing – original draft:** Madhulika Rai, Sarah M. Carter, Geetanjali Chawla, Jason M. Tennessen.

**Writing – review & editing:** Madhulika Rai, Sarah M. Carter, Geetanjali Chawla, Jason M. Tennessen.

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
