## [Decision Letter · Decision Letter 0]

4 Sep 2023

PONE-D-23-18419Characterization of genetic and molecular tools for studying the endogenous expression of Lactate dehydrogenase in Drosophila melanogasterPLOS ONE

Dear Dr. Tennessen,

Thank you for submitting your manuscript to PLOS ONE. After careful consideration, we feel that it has merit but does not fully meet PLOS ONE’s publication criteria as it currently stands. Therefore, we invite you to submit a revised version of the manuscript that addresses the points raised during the review process.

We look forward to receiving your revised manuscript.

Kind regards,

Jyotshna Kanungo, Ph.D.

Academic Editor

PLOS ONE

“We thank the Bloomington Drosophila Stock Center (NIH P40OD018537) for providing fly stocks, the Drosophila Genomics Resource Center (NIH 2P40OD010949) for genomic reagents, Flybase (NIH 5U41HG000739), and the Indiana University Light Microscopy Imaging Center. Targeted GC-MS analysis was conducted using instruments housed in the Indiana University Mass Spectrometry Facility, which is supported, in part, by NSF MRI Award 1726633. G.C is supported by the DBT/Wellcome Trust India Alliance Fellowship/Grant [grant number IA/I(S)/17/1/503085]. J.M.T. is supported by the National Institute of General Medical Sciences of the National Institutes of Health under a R35 Maximizing Investigators’ Research Award (MIRA; 1R35GM119557).”

“G.C is supported by the DBT/Wellcome Trust India Alliance Fellowship/Grant [grant number IA/I(S)/17/1/503085]. https://www.indiaalliance.org/fellowshiptype/basic-biomedical-research-fellowships. The funders had no role in study design, data collection and analysis, decision to publish, or preparation of the manuscript.

J.M.T. is supported by the National Institute of General Medical Sciences of the National Institutes of Health under a R35 Maximizing Investigators’ Research Award (MIRA; 1R35GM119557). https://nigms.nih.gov. The funders had no role in study design, data collection and analysis, decision to publish, or preparation of the manuscript.”

3.Please review your reference list to ensure that it is complete and correct. If you have cited papers that have been retracted, please include the rationale for doing so in the manuscript text, or remove these references and replace them with relevant current references. Any changes to the reference list should be mentioned in the rebuttal letter that accompanies your revised manuscript. If you need to cite a retracted article, indicate the article’s retracted status in the References list and also include a citation and full reference for the retraction notice.

Additional Editor Comments:

The reviewers have suggested minor revisions to the manuscript. Need for additional statistical analyses, references, and images for supplementary data has been raised. Certain aspects of the materials and methods section also need detailing.

Reviewers' comments:

Reviewer's Responses to Questions

**Comments to the Author**

1. Is the manuscript technically sound, and do the data support the conclusions?

Reviewer #1: Yes

Reviewer #2: Yes

2. Has the statistical analysis been performed appropriately and rigorously? 

Reviewer #1: I Don't Know

Reviewer #2: Yes

3. Have the authors made all data underlying the findings in their manuscript fully available?

Reviewer #1: No

Reviewer #2: Yes

4. Is the manuscript presented in an intelligible fashion and written in standard English?

Reviewer #1: Yes

Reviewer #2: Yes

5. Review Comments to the Author

Reviewer #1: In this manuscript Rai et. al. demonstrate where Ldh is localized in Drosophila larva across developmental stages and tissue types. The authors provide validation for valuable tools to examine expression of this enzyme that is highly relevant to development and many other physiological processes. Moreover, their discussion is clear and concise. The primary strengths of this study are the number of tissues measured and the direct comparison of similar tools in equivalent contexts. The weaknesses of this study are a lack of quantification for the fluorescence images across larval development, lack of any comment on individual variation, and absence of raw values for quantitative data with statistical analyses. Nonetheless, this study provides insightful findings that are suitable for publication in this journal. I recommend the editors accept this manuscript with revisions.

Here are my recommended revisions in order of importance:

1. Quantitative data provided in this manuscript is presented exclusively in Figure 1B-D. In the figure legend of Figure 1 the authors state that this data was analyzed using “one-way ANOVA followed by a post hoc Tukey HSD.” On page 9 the authors write in the methods section that for metabolite “Data were compared using a Kruskal–Walli’s test followed by a Dunn’s multiple comparison test.” The raw values and statistical analyses for all quantitative data (including the viability assay) were not provided. I recommend that the authors provide the raw values and statistical analyses (including test statistics and p value for all tests) for all quantitative data in the manuscript and clarify in the methods section which data were analyzed by ANOVA and Tukey or Kruskal-Wallis and Dunn’s.

2. In Figure 1 and S1, the authors provide qualitative data depicting detection of change in Ldh level across larval development using two different lines, Ldh-GFPGenomic and Ldh-mCherryGenomic. To strengthen these findings, I recommend the authors provide a quantification of the fluorescence depicted in these images in order to highlight the magnitude of changes that were detected and to compare more clearly to previous findings.

3. Throughout the manuscript, the authors provide images from a single representative individual animal fore every result. Although this may be sufficient for a binary measure whereby the outcome is only either positive or negative, instances whereby patterns are evaluated should be examined in more than a single animal. I recommend the authors provide supplementary images from additional animals for at least Figure 1, S1, and 2. At a minimum, even if additional images are not provided, I recommend that the authors discuss in the body text whether they observed any individual variation in their results from each tool they used, including Ldh-mCherry.

4. The authors highlight the discovery of new functions of Ldh on page 3. This description of Ldh seems to conflate the function of lactate to the function of Ldh. I recommend clarification of the distinction between functions attributed to Ldh and the substrates of Ldh, lactate and pyruvate, of which Ldh reversibly interconverts. Furthermore, a particularly important newly discovered function of lactate which warrants citing because it may have a significant impact on development is the ability of lactate to regulate cell cycle through zinc chelation (Liu, W., Wang, Y., Bozi, L.H.M., Fischer, P., Jedrychowski, M.P., Xiao, H., Wu, T., Darabedian, N., He, X., Mills, E.L., et al. (2023). Lactate regulates cell cycle by remodeling the anaphase promoting complex. Nature 616, 1–2. DOI:10.1038/s41586-023-05939-3 PMID:36921622.)

5. The authors state on page 3 “Ldh expression and activity are very high during this growth phase when compared to other phases of fly life-cycle [19]”. This citation only measures Ldh activity and not expression. I recommend adding a citation that provides evidence supporting the claimed changes in expression, for example (Graveley, B., Brooks, A., Carlson, J. et al. The developmental transcriptome of Drosophila melanogaster. Nature 471, 473–479 (2011). DOI: 10.1038/nature09715)

6. The authors cite three studies related to the connection between lactate metabolism activity and “neuronal health and aging [24-27]”. I recommend four additional critical citations (1. Hunt, L.C., and Demontis, F. (2021). Age-Related Increase in Lactate Dehydrogenase Activity in Skeletal Muscle Reduces Lifespan in Drosophila. J. Gerontol. A. Biol. Sci. Med. Sci. XX, 1–32. DOI:10.1093/gerona/glab260 PMID:34477202. 2. Frame, A.K., Robinson, J.W., Mahmoudzadeh, N.H., Tennessen, J.M., Simon, A.F., and Cumming, R.C. (2023). Aging and memory are altered by genetically manipulating lactate dehydrogenase in the neurons or glia of flies. Aging (Albany. NY). 10, 1–53. DOI:10.18632/aging.204565. 3. Long, D.M., Frame, A.K., Reardon, P.N., Cumming, R.C., Hendrix, D.A., Kretzschmar, D., and Giebultowicz, J.M. (2020). Lactate dehydrogenase expression modulates longevity and neurodegeneration in Drosophila melanogaster. Aging (Albany. NY). 12, 10041–10058. DOI:10.18632/aging.103373. 4. Lee, J.E., Oney, M., Frizzell, K., Phadnis, N., and Hollien, J. (2015). Drosophila melanogaster Activating Transcription Factor 4 Regulates Glycolysis During Endoplasmic Reticulum Stress. G3 Genes|Genomes|Genetics 5, 667–675. DOI:10.1534/g3.115.017269 PMID:25681259.

7. On page 6 the authors describe the production of Ldh-mCherryGenomic with reference to a past study which has used this resource and having produced this resource in a similar manner to Ldh-GFPGenomic. Citation 28 does not provide an explanation for the method used to generate Ldh-mCherryGenomic and citation 32 describes how Ldh-GFPGenomic was generated using a PCR based method and plasmid injection by Rainbow Transgenics. I recommend the authors clarify here whether the methods used for generating Ldh-mCherryGenomic were identical and cite Rainbow Transgenics if their services were used.

8. In the methods section the authors do not provide an explanation for how Ldh-mCherryGenomic was imaged. Was an antibody against mCherry required? I recommend the authors provide this information.

9. On page 8 the authors describe their confocal microscopy method. I recommend the authors provide the excitation and emission wavelengths used for detection of each fluorophore imaged, the method used for combining Z-stacks (e.g. maximum intensity projection vs average intensity projection), and the objectives used.

10. The authors describe the results from Figure one on page 10 without reference to Figure 1A or 1B. I recommend that the authors refer to those figures where they have described them.

11. On page 10 the authors acknowledge a slight decrease in lactate and 2HG in their Ldh-GFPGenomic rescue compared to control without reference to any statistical test in the figure or results. I recommend the authors provide the results of a statistical test comparing the rescue with control. Moreover, I recommend the authors add discussion of potential reasons for differences in fusion protein activity compared to endogenous Ldh. For example, protein metabolite interaction (Hicks, K.G., Cluntun, A.A., Schubert, H.L., Hackett, S.R., Berg, J.A., Leonard, P.G., Ajalla Aleixo, M.A., Zhou, Y., Bott, A.J., Salvatore, S.R., et al. (2023). Protein-metabolite interactomics of carbohydrate metabolism reveal regulation of lactate dehydrogenase. Science (80-. ). 379, 996–1003. DOI:10.1126/science.abm3452 PMID:36893255.), post-translational modification (Storey, K.B. (2016). Comparative enzymology—new insights from studies of an “old” enzyme, lactate dehydrogenase. Comp. Biochem. Physiol. Part - B Biochem. Mol. Biol. 199, 13–20. DOI:10.1016/j.cbpb.2015.12.004 PMID:26688543.), or post-transcriptional regulation (Jungmann, R.A., Huang, D., and Tian, D. (1998). Regulation of LDH-A gene expression by transcriptional and posttranscriptional signal transduction mechanisms. J. Exp. Zool. 282, 188–195. DOI:10.1002/(SICI)1097-010X(199809/10)282:1/2<188::AID-JEZ21>3.0.CO;2-P PMID:9723176.).

12. In Figure 2 the image for anti-Ldh staining without the entire ventral nerve cord visible. For fair comparison, I recommend addition of an image with anti-Ldh antibody with the ventral nerve chord entirely in frame like those larval nervous systems depicted for the GFP genomic and enhancer lines.

13. On page 13 the authors summarize their findings. For clarity and to highlight differences between the methods, I recommend the authors provide a summary table comparing the methodology and Ldh characteristics identified for the different methods they recommend. Moreover, I recommend the authors comment on the utility (or lack thereof) of the Ldh-mCherry line despite having characterized it less than the other methods. If the Ldh-mCherry line does not require any antibody staining to detect the signal above autofluorescence, then this may be an advantage for studies utilizing this line outside of the context whereby the authors have identified discrepant patterns of expression (pupal development).

14. In the legend for Figure 2, 3, 4, and 5 the authors state “For Ldh-GFPGenomic and Ldh-GFPenhancer expression analysis, the CNS was fixed and stained with an anti-GFP antibody as described in the methods.” Reference to the Ldh-GFPenhancer line in Figure 3 seems to be irrelevant because the authors also state in that figure legend that only Ldh-GFPgenomic was used. In addition, having fixation and staining procedures in the methods section makes mention of this method in the figure legends redundant unless they differ from the description in the methods section. I recommend that the authors remove all statements “For Ldh-GFPGenomic and Ldh-GFPenhancer expression analysis, the CNS was fixed and stained with an anti-GFP antibody as described in the methods.“ from the figure legend unless there is a reason to distinguish a method in any figure from the method described in the methods section.

15. For all Figures, the scale bars and labels should be formatted consistently. I recommend that the authors label staining for GFP in Ldh-GFPGenomic and Ldh-GFPEnhancer in Figure 2, 3, 4, 5, and 6 consistently as either only αGFP or only LDH. I recommend the authors make all labels denoting the stain used in images below consistently displayed in a font color matching the pseudocolor in the image below, as they have done in Figure 3. I recommend that the authors remove all labels describing the length of scale bars from within the image and keep this description of scale bar length restricted to the Figure legend, as they have done in Figure 5.

16. In the legend for Figure 3, the authors state refer to (I-K) whereby they seem to be referring to (I-L). I recommend the authors change this to (I-L).

17. The label above Figure 5F and L is (αGFP). This seems to be a mistake. If this is incorrect, I recommend the authors change this to αLDH as they have done in Figure 4.

18. In the legend for Figure S2 the authors state that the brightness was increased in panel L. I recommend the authors provide the percentage the brightness was increased compared to the other images in this figure.

19. On page 13 the statement “Previous studies of the Ldh enzyme activity suggest that Ldh expression levels are relatively low in the larval salivary glands and fat body” is not cited. I recommend the authors add a citation supporting this claim.

20. The authors state “(see methods)” on page 5 where reiterating the catalog number and supplier of the antibody here would provide more clarity to the reader: (Bosterbio; DZ41222)

Reviewer #2: Dear Editor,

In the manuscript "Characterization of genetic and molecular tools for studying the endogenous expression of Lactate dehydrogenase in Drosophila melanogaster", Ray et al. provide a detailed and accurate characterization of D. melanogaster LDH reporter lines and a new anti-LDH antibody.

Although the information provided in this manuscript may be limited to a specialized audience, the experiments reported here provide an important set of data to validate l reagents previously used in several publications and now available to the Drosophila community.

I have a few minor comments below that I think should be addressed prior to publication, but overall I consider the manuscript to be of high quality and have no major comments that would preclude its publication after this minor corrections.

Sincerely

Minor comments:

- Figure 1C: The figure legend indicates that the Ldh-GFP fusion protein is able to restore steady-state levels of (C) lactate and (D) 2-hydroxyglutarate (2HG).

Although lactate and 2HG levels are significantly increased in p{Ldh-GFP} compared to Ldh null mutants, the statistical difference in metabolite levels between Ldh and p{Ldh-GFP} was not tested. The phrase "restore steady-state levels" is ambiguous in this context. The authors should either perform an additional statistical test and retain the sentence only in case of lack of statistical differences or rephrase this sentence to more accurately describe the observed results, as is already the case in the Results section.

- Figure 3C: References justifying the choice of different markers (REPO, PROS, MIRA) could be added to the text.

- The term "AMP cluster" should be defined.

6. PLOS authors have the option to publish the peer review history of their article (what does this mean?). If published, this will include your full peer review and any attached files.

Reviewer #1: **Yes: **Ariel K. Frame

Reviewer #2: No

---

## [Author Response · Author response to Decision Letter 0]

29 Nov 2023

Reviewer #1: In this manuscript Rai et. al. demonstrate where Ldh is localized in Drosophila larva across developmental stages and tissue types. The authors provide validation for valuable tools to examine expression of this enzyme that is highly relevant to development and many other physiological processes. Moreover, their discussion is clear and concise. The primary strengths of this study are the number of tissues measured and the direct comparison of similar tools in equivalent contexts. The weaknesses of this study are a lack of quantification for the fluorescence images across larval development, lack of any comment on individual variation, and absence of raw values for quantitative data with statistical analyses. Nonetheless, this study provides insightful findings that are suitable for publication in this journal. I recommend the editors accept this manuscript with revisions.

We thank the reviewer for his helpful and insightful comments.

Here are my recommended revisions in order of importance:

1. Quantitative data provided in this manuscript is presented exclusively in Figure 1B-D. In the figure legend of Figure 1 the authors state that this data was analyzed using “one-way ANOVA followed by a post hoc Tukey HSD.” On page 9 the authors write in the methods section that for metabolite “Data were compared using a Kruskal–Walli’s test followed by a Dunn’s multiple comparison test.” The raw values and statistical analyses for all quantitative data (including the viability assay) were not provided. I recommend that the authors provide the raw values and statistical analyses (including test statistics and p value for all tests) for all quantitative data in the manuscript and clarify in the methods section which data were analyzed by ANOVA and Tukey or Kruskal-Wallis and Dunn’s.

Thank you for noticing the discrepancy in descriptions of statistical test – Kruskal-Wallis and Dunn’s was used in Figure 1 – the text now accurately describes the test used. Data for figure panels 1B-D are now included in Table S1, as is the statistical analysis. 

2. In Figure 1 and S1, the authors provide qualitative data depicting detection of change in Ldh level across larval development using two different lines, Ldh-GFPGenomic and Ldh-mCherryGenomic. To strengthen these findings, I recommend the authors provide a quantification of the fluorescence depicted in these images in order to highlight the magnitude of changes that were detected and to compare more clearly to previous findings.

We appreciate the reviewer’s suggestions, however, we believe this level analysis is unnecessary. Ldh expression levels have been repeatedly quantified during larval development and our images demonstrate that Ldh-GFP qualitatively exhibits a similar expression pattern. Since the purpose of our study is to validate reagents for studying Ldh expression, we believe that such an analysis is best conducted in a future study focused on the mechanisms that temporally regulate Ldh expression.

3. Throughout the manuscript, the authors provide images from a single representative individual animal for every result. Although this may be sufficient for a binary measure whereby the outcome is only either positive or negative, instances whereby patterns are evaluated should be examined in more than a single animal. I recommend the authors provide supplementary images from additional animals for at least Figure 1, S1, and 2. At a minimum, even if additional images are not provided, I recommend that the authors discuss in the body text whether they observed any individual variation in their results from each tool they used, including Ldh-mCherry.

We completely agree. As noted in the text, we only described observations that were similar or identical across all three reagents (see Lines 267-270). And while we only presented representative images, the fact that we included images from three independent reagents for nearly all tissues far exceeds the current standards of the developmental biology community. We would also note that, in many cases, the observations were largely binary. For example, neuroblasts vs GMCs, stellate cells vs principal cells, and AMPs vs enterocytes. We’ve added additional text to emphasize this point. 

Regarding the Ldh-mCherry fusion, we decided to not characterize the expression pattern on this reagent because the fusion protein appears to be excessively stabilized. We’ve added a few sentences noting that this reagent should be used with caution when quantifying Ldh expression levels.

4. The authors highlight the discovery of new functions of Ldh on page 3. This description of Ldh seems to conflate the function of lactate to the function of Ldh. I recommend clarification of the distinction between functions attributed to Ldh and the substrates of Ldh, lactate and pyruvate, of which Ldh reversibly interconverts. Furthermore, a particularly important newly discovered function of lactate which warrants citing because it may have a significant impact on development is the ability of lactate to regulate cell cycle through zinc chelation (Liu, W., Wang, Y., Bozi, L.H.M., Fischer, P., Jedrychowski, M.P., Xiao, H., Wu, T., Darabedian, N., He, X., Mills, E.L., et al. (2023). Lactate regulates cell cycle by remodeling the anaphase promoting complex. Nature 616, 1–2. DOI:10.1038/s41586-023-05939-3 PMID:36921622.)

We’ve added text to Line 63 to address this issue. We have also added the suggested reference.

5. The authors state on page 3 “Ldh expression and activity are very high during this growth phase when compared to other phases of fly life-cycle [19]”. This citation only measures Ldh activity and not expression. I recommend adding a citation that provides evidence supporting the claimed changes in expression, for example (Graveley, B., Brooks, A., Carlson, J. et al. The developmental transcriptome of Drosophila melanogaster. Nature 471, 473–479 (2011). DOI: 10.1038/nature09715)

Thank you for noticing the missing reference. The appropriate reference for this statement is Abu-Shumays RL, Fristrom JW. IMP-L3, A 20-hydroxyecdysone-responsive gene encodes Drosophila lactate dehydrogenase: structural characterization and developmental studies. Dev Genet. 1997;20(1):11-22. doi: 10.1002/(SICI)1520-6408(1997)20:1<11::AID-DVG2>3.0.CO;2-C. PMID: 9094207, not Gravely et. al. We have added this reference to the text (Line 80).

6. The authors cite three studies related to the connection between lactate metabolism activity and “neuronal health and aging [24-27]”. I recommend four additional critical citations (1. Hunt, L.C., and Demontis, F. (2021). Age-Related Increase in Lactate Dehydrogenase Activity in Skeletal Muscle Reduces Lifespan in Drosophila. J. Gerontol. A. Biol. Sci. Med. Sci. XX, 1–32. DOI:10.1093/gerona/glab260 PMID:34477202. 2. Frame, A.K., Robinson, J.W., Mahmoudzadeh, N.H., Tennessen, J.M., Simon, A.F., and Cumming, R.C. (2023). Aging and memory are altered by genetically manipulating lactate dehydrogenase in the neurons or glia of flies. Aging (Albany. NY). 10, 1–53. DOI:10.18632/aging.204565. 3. Long, D.M., Frame, A.K., Reardon, P.N., Cumming, R.C., Hendrix, D.A., Kretzschmar, D., and Giebultowicz, J.M. (2020). Lactate dehydrogenase expression modulates longevity and neurodegeneration in Drosophila melanogaster. Aging (Albany. NY). 12, 10041–10058. DOI:10.18632/aging.103373. 4. Lee, J.E., Oney, M., Frizzell, K., Phadnis, N., and Hollien, J. (2015). Drosophila melanogaster Activating Transcription Factor 4 Regulates Glycolysis During Endoplasmic Reticulum Stress. G3 Genes|Genomes|Genetics 5, 667–675. DOI:10.1534/g3.115.017269 PMID:25681259.

Thank you for suggesting these citations – Frame et al was in the near final version of the manuscript and was somehow removed. We have added the additional suggested references to our manuscript.

7. On page 6 the authors describe the production of Ldh-mCherryGenomic with reference to a past study which has used this resource and having produced this resource in a similar manner to Ldh-GFPGenomic. Citation 28 does not provide an explanation for the method used to generate Ldh-mCherryGenomic and citation 32 describes how Ldh-GFPGenomic was generated using a PCR based method and plasmid injection by Rainbow Transgenics. I recommend the authors clarify here whether the methods used for generating Ldh-mCherryGenomic were identical and cite Rainbow Transgenics if their services were used.

We feel that the methods section is clear that the Ldh-mCherry[Genomic] construct was generated using identical methods to the Ldh-GFP[Genomic], and a description of how Ldh-GFP[Genomic] was generated is available in the cited reference.

8. In the methods section the authors do not provide an explanation for how Ldh-mCherryGenomic was imaged. Was an antibody against mCherry required? I recommend the authors provide this information.

These images are from living larvae using a fluorescent microscope without fixation. We have added a new section “Whole larvae imaging” under the methods section. Lines 153-159.

9. On page 8 the authors describe their confocal microscopy method. I recommend the authors provide the excitation and emission wavelengths used for detection of each fluorophore imaged, the method used for combining Z-stacks (e.g. maximum intensity projection vs average intensity projection), and the objectives used.

The relevant information is now included in the Methods section (Line 198-200). 

10. The authors describe the results from Figure one on page 10 without reference to Figure 1A or 1B. I recommend that the authors refer to those figures where they have described them.

We have added the Figure 1A and 1B in the text. Thank you for catching the mistake.

11. On page 10 the authors acknowledge a slight decrease in lactate and 2HG in their Ldh-GFPGenomic rescue compared to control without reference to any statistical test in the figure or results. I recommend the authors provide the results of a statistical test comparing the rescue with control. Moreover, I recommend the authors add discussion of potential reasons for differences in fusion protein activity compared to endogenous Ldh. For example, protein metabolite interaction (Hicks, K.G., Cluntun, A.A., Schubert, H.L., Hackett, S.R., Berg, J.A., Leonard, P.G., Ajalla Aleixo, M.A., Zhou, Y., Bott, A.J., Salvatore, S.R., et al. (2023). Protein-metabolite interactomics of carbohydrate metabolism reveal regulation of lactate dehydrogenase. Science (80-. ). 379, 996–1003. DOI:10.1126/science.abm3452 PMID:36893255.), post-translational modification (Storey, K.B. (2016). Comparative enzymology—new insights from studies of an “old” enzyme, lactate dehydrogenase. Comp. Biochem. Physiol. Part - B Biochem. Mol. Biol. 199, 13–20. DOI:10.1016/j.cbpb.2015.12.004 PMID:26688543.), or post-transcriptional regulation (Jungmann, R.A., Huang, D., and Tian, D. (1998). Regulation of LDH-A gene expression by transcriptional and posttranscriptional signal transduction mechanisms. J. Exp. Zool. 282, 188–195. DOI:10.1002/(SICI)1097-010X(199809/10)282:1/2<188::AID-JEZ21>3.0.CO;2-P PMID:9723176.).

The data and statistical results can now be found in Table S1. Appropriate references regarding the manner by which Ldh functions as a tetramer have also been added (Lines 231-233).

12. In Figure 2 the image for anti-Ldh staining without the entire ventral nerve cord visible. For fair comparison, I recommend addition of an image with anti-Ldh antibody with the ventral nerve chord entirely in frame like those larval nervous systems depicted for the GFP genomic and enhancer lines.

We appreciate the suggestion, but have decided to not replace this image.

13. On page 13 the authors summarize their findings. For clarity and to highlight differences between the methods, I recommend the authors provide a summary table comparing the methodology and Ldh characteristics identified for the different methods they recommend. Moreover, I recommend the authors comment on the utility (or lack thereof) of the Ldh-mCherry line despite having characterized it less than the other methods. If the Ldh-mCherry line does not require any antibody staining to detect the signal above autofluorescence, then this may be an advantage for studies utilizing this line outside of the context whereby the authors have identified discrepant patterns of expression (pupal development).

Thank you for the suggestion, but we decided not at add a summary table. We have added addition text about the use of the mCherry reagent (see Lines 329-333).

14. In the legend for Figure 2, 3, 4, and 5 the authors state “For Ldh-GFPGenomic and Ldh-GFPenhancer expression analysis, the CNS was fixed and stained with an anti-GFP antibody as described in the methods.” Reference to the Ldh-GFPenhancer line in Figure 3 seems to be irrelevant because the authors also state in that figure legend that only Ldh-GFPgenomic was used. In addition, having fixation and staining procedures in the methods section makes mention of this method in the figure legends redundant unless they differ from the description in the methods section. I recommend that the authors remove all statements “For Ldh-GFPGenomic and Ldh-GFPenhancer expression analysis, the CNS was fixed and stained with an anti-GFP antibody as described in the methods.“ from the figure legend unless there is a reason to distinguish a method in any figure from the method described in the methods section.

We have removed the text from the figure legends as asked by the reviewer.

15. For all Figures, the scale bars and labels should be formatted consistently. I recommend that the authors label staining for GFP in Ldh-GFPGenomic and Ldh-GFPEnhancer in Figure 2, 3, 4, 5, and 6 consistently as either only αGFP or only LDH. I recommend the authors make all labels denoting the stain used in images below consistently displayed in a font color matching the pseudocolor in the image below, as they have done in Figure 3. I recommend that the authors remove all labels describing the length of scale bars from within the image and keep this description of scale bar length restricted to the Figure legend, as they have done in Figure 5.

We have made the recommended changes in the figures.

16. In the legend for Figure 3, the authors state refer to (I-K) whereby they seem to be referring to (I-L). I recommend the authors change this to (I-L).

We have corrected this in the manuscript.

17. The label above Figure 5F and L is (αGFP). This seems to be a mistake. If this is incorrect, I recommend the authors change this to αLDH as they have done in Figure 4.

We have corrected Figure 5F.

18. In the legend for Figure S2 the authors state that the brightness was increased in panel L. I recommend the authors provide the percentage the brightness was increased compared to the other images in this figure.

We’ve added this information to the figure legend.

19. On page 13 the statement “Previous studies of the Ldh enzyme activity suggest that Ldh expression levels are relatively low in the larval salivary glands and fat body” is not cited. I recommend the authors add a citation supporting this claim.

Thank you for noticing this omission. The references seem to have been lost in the final version of the manuscript. We have added back the appropriate references.

20. The authors state “(see methods)” on page 5 where reiterating the catalog number and supplier of the antibody here would provide more clarity to the reader: (Bosterbio; DZ41222)

We have made this change to our manuscript.

Reviewer #2: Dear Editor,

In the manuscript "Characterization of genetic and molecular tools for studying the endogenous expression of Lactate dehydrogenase in Drosophila melanogaster", Ray et al. provide a detailed and accurate characterization of D. melanogaster LDH reporter lines and a new anti-LDH antibody.

Although the information provided in this manuscript may be limited to a specialized audience, the experiments reported here provide an important set of data to validate l reagents previously used in several publications and now available to the Drosophila community.

I have a few minor comments below that I think should be addressed prior to publication, but overall I consider the manuscript to be of high quality and have no major comments that would preclude its publication after this minor corrections.

Sincerely

We thank the reviewer for the kind and helpful overview of our paper!

Minor comments:

- Figure 1C: The figure legend indicates that the Ldh-GFP fusion protein is able to restore steady-state levels of (C) lactate and (D) 2-hydroxyglutarate (2HG).

Although lactate and 2HG levels are significantly increased in p{Ldh-GFP} compared to Ldh null mutants, the statistical difference in metabolite levels between Ldh and p{Ldh-GFP} was not tested. The phrase "restore steady-state levels" is ambiguous in this context. The authors should either perform an additional statistical test and retain the sentence only in case of lack of statistical differences or rephrase this sentence to more accurately describe the observed results, as is already the case in the Results section.

We agree with the reviewer and have performed additional statistical tests. We also rephrased the figure legend for Figure 1C and D and have included the raw data for these figure panels, as well as the statistical test information, in Table S1.

- Figure 3C: References justifying the choice of different markers (REPO, PROS, MIRA) could be added to the text.

We agree and have added references to the text justifying the choice of the different markers.

- The term "AMP cluster" should be defined.

We have defined the term “AMP cluster” in the manuscript and added the appropriate references (Lines 288-292). Thank you for the suggestion.

---

## [Decision Letter · Decision Letter 1]

12 Dec 2023

PONE-D-23-18419R1Characterization of genetic and molecular tools for studying the endogenous expression of Lactate dehydrogenase in Drosophila melanogasterPLOS ONE

Dear Dr. Tennessen,

Thank you for submitting your manuscript to PLOS ONE. After careful consideration, we feel that it has merit but does not fully meet PLOS ONE’s publication criteria as it currently stands. Therefore, we invite you to submit a revised version of the manuscript that addresses the points raised during the review process. A reviewer has suggested minor revision that includes consistent labeling of figures.

We look forward to receiving your revised manuscript.

Kind regards,

Jyotshna Kanungo, Ph.D.

Academic Editor

PLOS ONE

Journal Requirements:

Reviewers' comments:

Reviewer's Responses to Questions

**Comments to the Author**

1. If the authors have adequately addressed your comments raised in a previous round of review and you feel that this manuscript is now acceptable for publication, you may indicate that here to bypass the “Comments to the Author” section, enter your conflict of interest statement in the “Confidential to Editor” section, and submit your "Accept" recommendation.

Reviewer #1: All comments have been addressed

Reviewer #2: All comments have been addressed

2. Is the manuscript technically sound, and do the data support the conclusions?

Reviewer #1: Yes

Reviewer #2: Yes

3. Has the statistical analysis been performed appropriately and rigorously? 

Reviewer #1: Yes

Reviewer #2: Yes

4. Have the authors made all data underlying the findings in their manuscript fully available?

Reviewer #1: Yes

Reviewer #2: Yes

5. Is the manuscript presented in an intelligible fashion and written in standard English?

Reviewer #1: Yes

Reviewer #2: Yes

6. Review Comments to the Author

Reviewer #1: All comments were adequately addressed. There are a few very minor changes I recommend before final publication:

1) I agree that Abu-Shumays et al. is sufficient evidence for the claim that Ldh expression is relatively high during larval developement. But, it seems this citation was added to the sentence prior to this claim (Line 79). I recommend moving it to the sentence discussion Ldh expression (Line 80).

2) The method for combining Z-stacks is still not provided (e.g. maximum intensity projection / average intensity projection). I recommend this is added to the sentence on Line 196.

3) Figure 4 and 5 have the green channel legend labeled αLDH whereas figure 2 and everywhere else in the body text and figure legends denotes this without all uppercase as αLdh. I recommend changing figure 4 and 5 to match everywhere else (i.e. αLdh).

Reviewer #2: I consider that the authors have responded adequately to all my comments. I recommend the publication of this study.

7. PLOS authors have the option to publish the peer review history of their article (what does this mean?). If published, this will include your full peer review and any attached files.

Reviewer #1: **Yes: **Ariel K. Frame

Reviewer #2: No

---

## [Author Response · Author response to Decision Letter 1]

13 Dec 2023

Response to Reviewers Comments:

Reviewer #1: All comments were adequately addressed. There are a few very minor changes I recommend before final publication:

1) I agree that Abu-Shumays et al. is sufficient evidence for the claim that Ldh expression is relatively high during larval developement. But, it seems this citation was added to the sentence prior to this claim (Line 79). I recommend moving it to the sentence discussion Ldh expression (Line 80).

Done.

2) The method for combining Z-stacks is still not provided (e.g. maximum intensity projection / average intensity projection). I recommend this is added to the sentence on Line 196.

We appreciate the suggestion, but would kindly correct the reviewer and note that the methods sections is accurate and does not need edits. As already stated in the text, we did not combine Z-stacks in the figure, but rather used a representative section.

3) Figure 4 and 5 have the green channel legend labeled αLDH whereas figure 2 and everywhere else in the body text and figure legends denotes this without all uppercase as αLdh. I recommend changing figure 4 and 5 to match everywhere else (i.e. αLdh).

Done.

Reviewer #2: I consider that the authors have responded adequately to all my comments. I recommend the publication of this study.

Thank you!

---

## [Decision Letter · Decision Letter 2]

18 Dec 2023

Characterization of genetic and molecular tools for studying the endogenous expression of Lactate dehydrogenase in Drosophila melanogaster

PONE-D-23-18419R2

Dear Dr. Tennessen,

We’re pleased to inform you that your manuscript has been judged scientifically suitable for publication and will be formally accepted for publication once it meets all outstanding technical requirements.

Kind regards,

Jyotshna Kanungo, Ph.D.

Academic Editor

PLOS ONE

Additional Editor Comments (optional):

Reviewers' comments:

Reviewer's Responses to Questions

**Comments to the Author**

1. If the authors have adequately addressed your comments raised in a previous round of review and you feel that this manuscript is now acceptable for publication, you may indicate that here to bypass the “Comments to the Author” section, enter your conflict of interest statement in the “Confidential to Editor” section, and submit your "Accept" recommendation.

Reviewer #1: All comments have been addressed

2. Is the manuscript technically sound, and do the data support the conclusions?

Reviewer #1: Yes

3. Has the statistical analysis been performed appropriately and rigorously? 

Reviewer #1: Yes

4. Have the authors made all data underlying the findings in their manuscript fully available?

Reviewer #1: Yes

5. Is the manuscript presented in an intelligible fashion and written in standard English?

Reviewer #1: Yes

6. Review Comments to the Author

Reviewer #1: (No Response)

7. PLOS authors have the option to publish the peer review history of their article (what does this mean?). If published, this will include your full peer review and any attached files.

Reviewer #1: **Yes: **Ariel K. Frame

---

## [Editor Report · Acceptance letter]

21 Dec 2023

PONE-D-23-18419R2 

PLOS ONE

Dear Dr. Tennessen, 

I'm pleased to inform you that your manuscript has been deemed suitable for publication in PLOS ONE. Congratulations! Your manuscript is now being handed over to our production team.

Kind regards, 

on behalf of

Dr. Jyotshna Kanungo 

Academic Editor

PLOS ONE